# Carotenoids Biosynthesis, Accumulation, and Applications of a Model Microalga *Euglena*
*gracilis*

**DOI:** 10.3390/md20080496

**Published:** 2022-07-31

**Authors:** Rao Yao, Wen Fu, Ming Du, Zi-Xi Chen, An-Ping Lei, Jiang-Xin Wang

**Affiliations:** Shenzhen Key Laboratory of Marine Bioresource and Eco-Environmental Science, Shenzhen Engineering Laboratory for Marine Algal Biotechnology, Guangdong Provincial Key Laboratory for Plant Epigenetics, College of Life Sciences and Oceanography, Shenzhen University, Shenzhen 518060, China; yaorao2021@email.szu.edu.cn (R.Y.); fuwen2021@email.szu.edu.cn (W.F.); duming2020@email.szu.edu.cn (M.D.); chenzx@szu.edu.cn (Z.-X.C.)

**Keywords:** *Euglena*, carotenoids, biosynthesis, applications, accumulation

## Abstract

The carotenoids, including lycopene, lutein, astaxanthin, and zeaxanthin belong to the isoprenoids, whose basic structure is made up of eight isoprene units, resulting in a C40 backbone, though some of them are only trace components in *Euglena*. They are essential to all photosynthetic organisms due to their superior photoprotective and antioxidant properties. Their dietary functions decrease the risk of breast, cervical, vaginal, and colorectal cancers and cardiovascular and eye diseases. Antioxidant functions of carotenoids are based on mechanisms such as quenching free radicals, mitigating damage from reactive oxidant species, and hindering lipid peroxidation. With the development of carotenoid studies, their distribution, functions, and composition have been identified in microalgae and higher plants. Although bleached or achlorophyllous mutants of *Euglena* were among the earliest carotenoid-related microalgae under investigation, current knowledge on the composition and biosynthesis of these compounds in *Euglena* is still elusive. This review aims to overview what is known about carotenoid metabolism in *Euglena*, focusing on the carotenoid distribution and structure, biosynthesis pathway, and accumulation in *Euglena* strains and mutants under environmental stresses and different culture conditions. Moreover, we also summarize the potential applications in therapy preventing carcinogenesis, cosmetic industries, food industries, and animal feed.

## 1. Introduction

*Euglena*, a unicellular flagellate, has been studied for over 100 years by researchers worldwide. As a potential candidate for new functional food resources, it contains many high-value products, including pigments, essential proteins, vitamins, unsaturated fatty acids, a unique β-1,3-glucan called paramylon, and other valuable compounds [1]. Carotenoids are useful pigments in fungi, algae, plants, some photosynthetic bacteria, and animals. They play crucial roles in light-harvesting and protecting cells from photooxidative damage in photosynthetic organisms [2]. Based on chemical structures, carotenoids are often divided into carotenes and xanthophylls. In the first group, it contains α-carotene, β-carotene, and γ-carotene. The other group contains oxygen atoms: lutein, zeaxanthin, fucoxanthin, and peridinin [3]. Although more and more studies have been done on carotenoids of some microalgae, such as *Chlorella* and *Haematococcus*, due to their commercial values, we still have many questions instead of answers about carotenoids in *Euglena*. As we all know, some microalgae carotenoid biosynthesis pathways have been investigated, such as *Chlamy**domonas*, *Haematococcus*, and cyanobacteria [4]. According to recent studies, the synthesis pathway of carotenoids has not been fully identified in *Euglena*. In addition, environmental factors such as heavy metals, pH, temperature, and so on can also affect carotenoid accumulation in *Euglena*. After 16 h of artificial UV-B irradiation, the accumulation of astaxanthin diester in cells is much more than without that radiation [5]. The stigma composed of carotenoids in *Euglena* as a functional device is related to phototaxis [6]. There are two major hypotheses about the function of eyespot in *Euglena*, including, (*i*) as a device of shading, which could help the cell to change its orientation when the direction of light changes; (*ii*) as a device of primary photoreceptor [7]. According to the study by Kato in 2020, the eyespot is essential for the phototaxis of Euglenophyta, and the carotenoids in the eyespot apparatus perform an important role in photoreception [8]. 

Traditionally, carotenoids as natural color enhancers have been used in the food and feed industries. For instance, the carotenoids are added to the cultivation of salmon, eggs, and chicken meat as a food color [9]. In addition, the health and fertility of grain-fed cattle could be improved by adding β-carotene [10]. Antioxidant functions of carotenoids are based on mechanisms such as quenching free radicals, mitigating damage from reactive oxidant species, and hindering lipid peroxidation. Moreover, with the development of fundamental research, the functions of antioxidants and anticancer come to our eyes. Their dietary functions decrease the risk of breast, cervical, vaginal, and colorectal cancers, and cardiovascular and eye diseases. 

With the development of carotenoid studies, their distribution, functions, and composition have been identified in microalgae and higher plants. Although bleached or achlorophyllous mutants of *Euglena* were among the earliest carotenoid-related microalgae under investigation, current knowledge on the composition and biosynthesis of these compounds in *Euglena* is still elusive. This review aims to collect and analyze the knowledge of carotenoids in *Euglena*, focusing on the carotenoid distribution and structure, biosynthesis, and accumulation in *Euglena* strains and mutants under different cultivation conditions. Moreover, we also summarize the potential applications in therapy preventing carcinogenesis, cosmetic industries, food industries, and animal feed.

## 2. Composition and Structures of Carotenoids in *Euglena*

Carotenoids are essential pigments for photosynthesis in microalgae and higher plants. With the development of analytical instruments, for instance, high performance liquid chromatography (HPLC), mass spectrometric (MS), and nuclear magnetic resonance (NMR), new structural carotenoids can be successfully determined [11]. The structure and distribution of significant carotenoids in *Euglena* are illustrated in Figure 1 and Table 1, respectively. The major carotenoids were found in *E. gracilis,* including diadinoxanthin, diatoxanthin, β-carotene, and neoxanthin, as shown in Table 1. The carotenoids in the stigma of *Euglena* were diadinoxanthin, diatoxanthin and β-carotene, which are related to phototaxis [12]. Although β-carotene was found in every species of *Euglena*, its content was not always the highest. β-carotene, with the highest provitamin A activity, was converted to retinol(vitamin A) *via* an oxidation process, which was catalyzed by enzyme β,β-carotene-15,15-monooxygenase [13]. β-carotene is present in reaction-center complexes and the light-harvesting complexes of the photosystem, which may have protective functions [4].

In 2014, the composition of carotenoids in *E**. sanguinea* was reported by Deli [14]. There are three kinds of pigments among carotenoids: diatoxanthin, lutein, and β-carotene.

Among them, the diatoxanthin is the highest pigment [14]. Astaxanthin is a commercial pigment popular among people worldwide as an antioxidant. *Haematoccus pluvialis*, the resource of astaxanthin, has become one of the most economical spices among algae [18]. Astaxanthin is also found in non-photosynthetic mutants of *E. gracilis*, suggesting the synthesis pathway of astaxanthin might be present in *E. gracilis* [19]. Astaxanthin was also detected in *Euglena sanguinea* [14] and *E. rubida* [15]. Astaxanthin, mutatoxanthin, and β-carotene were the top 3 pigments in *E rubida* by B. Czeczuga in 1974. The content of astaxanthin is higher than the rest [15]. Carotenes contain α-carotene, β-carotene, γ-carotene and lycopene.

## 3. Carotenoids Biosynthesis Pathways in *Euglena*

Carotenoids are synthesized in the plastids, such as chloroplasts and chromoplasts, in microalgae and plants [20]. The carotenoids biosynthesis pathway in microalgae is similar to that of streptophytes. However, the carotenoids biosynthesis pathway in microalgae is more complicated than that of terrestrial plants due to gene duplications, gene loss, gene transfer, etc. [21]. The details of the carotenoids biosynthesis pathway have been studied for many years in some microalgae, but *Euglena* still presents many unknowns. The carotenoid biosynthesis pathways are proposed according to the composition of carotenoids in *Euglena* instead of genetic evidence. The biosynthesis pathway of kinds of carotenoids and its derivative are shown in Figure 2. 

### 3.1. IDI to GGPP

The synthesis of isoprenoid isopentenyl pyrophosphate (IPP) is an irreplaceable process in carotenoid biosynthesis, IPP, a C5-compound, through two pathways. One way is the mevalonate pathway (MVA), and the other is the non-mevalonate pathway (MEP), which takes place in cytosol and plastids, respectively [22]. Only one of two pathways exists in one organism, or both are on different sides of the same cell. Firstly, the MVA pathway acts in all phototrophic algae with secondary plastids [23]. Furthermore, the MEP pathway is essential for all organisms that contain plastids, no matter the presence of oxygenic photosynthesis in cells [24]. However, compared with the algae which contain the DOXP/MEP pathway, such as *Chlorella*, *Chlamydomonas*, and *Scenedesmus*, the formation of cytosolic sterols and photo-synthetic isoprenoids in *Euglena* both occur via the MVA pathway [25]. The biosynthesis of IPP/DMAPP consists of six steps via the MVA pathway. In the first two steps, two acetyl-CoA molecules condense into an acetoacetyl-CoA catalyzed by acetoacetyl-CoA thiolase (AACT). Then, an acetyl-CoA is added to acetoacetyl-CoA catalyzed by 3-hydroxy-3-methylglutaryl-CoA synthase, and an HMG-CoA is generated. In the subsequent step, the HMG-CoA is converted to MVA, which must consume two NADPH molecules [21]. An MVA molecule generated the IPP through two-time phosphorylation and one ATP-coupled decarboxylation reaction. The MVA pathway involves seven enzymes, such as acetoacetyl-CoA thiolase (AACT), 3-hydroxy-3-methylglutaryl-CoA synthase (HMGS), 3-hydroxy-3-methylglutaryl-CoA reductase (HMGR), 5-phosphomevalonate kinase (PMK), mevalonate kinase (MVK), dimethylallyl diphosphate isomerase (IDI). In the final step, IPP is converted to DMAPP catalyzed by the IDI enzyme [26]. The overexpression of the IPP could increase the content of carotenoids. For instance, the accumulation of carotenoids has been enhanced with the overexpression of lycopeneβ-cyclase and β-carotene-C-4-oxygenase under high illumination [27].

Geranylgeranyl diphosphate (GGPP) is a metabolic predecessor of the biosynthesis of carotenoids [21]. As a precursor for phytoene, the GGPP is also essential for the carotenoid biosynthesis pathway. Geranylgeranyl diphosphate (GGPP) formation requires three steps (Figure 2). That process involves three types of enzymes, GPPS, FPPS, and GGPPS [28]. *EgcrtE* is a functional gene that encodes GGPP synthase in *E. gracilis* [29]. In the initial step, C10-compound was synthesized by adding one IPP and one DMAPP molecule. Additionally, C10-compound is converted to FPP by the FPPS enzyme. In the final step, GGPP is formed by adding one FPP and IPP.

### 3.2. GGPP to Lycopene

The first step of carotenoid biosynthesis is the formation of phytoene. Phytoene, C40-compound, a colorless pigment, is synthesized by condensing two GGPP molecules, which could be unutilized for nutraceutical and cosmetic products [30]. This process involves the enzyme phytoene synthase (PSY). PSY has a significant effect on the carotenoids biosynthesis pathway due to its rate-limiting and total flux controlling abilities, which decide the content of carotenoids [21]. *EgcrtB* was confirmed to encode phytoene synthase in *E. gracilis*. The content of carotenoids has a significant decrease in *EgcrtB*-suppressed cells [31].

The phytoene, a red color pigment, is converted to lycopene [32]. This process involves four desaturation steps and isomerization [2]. The whole process includes five steps and four conserved enzymes. In the first two-step, phytoene is converted to 9,15,9-tri-cis-ξ-carotene by the PDS enzyme. This compound is yellow. The phytoene desaturase genes (*EgcrtP1* and *EgcrtP2*) were identified by Kato in 2018, and both could encode the phytoene desaturase [33]. The 9,15,9-tri-cis-ξ-carotene then is converted to 9,9-dis-cis-ξ-carotene and then to prolycopene, and final to lycopene are catalyzed by Z-ISO, ZDS, and CrtlSO, respectively [21]. ζ-carotene isomerase gene (Z-ISO) in *E. gracilis* was related to Z-ISO of chlorophyta. What is more, Z-ISO is essential for oxygenic phototrophs, but not be found in anoxygenic phototrophs [33]. ζ-carotene desaturase gene (*EgcrtQ*) in *E. gracilis* has been proven to be a clade of ZDS [34]. Various studies reported that three genes encoding PDS, ZDS, and CRTISO, are present in all three groups of algae (Chlorophyta, Phaeophyta and Rhodophyta). However, the gene for Z-ISO is absent in red algae. There are no changes during carotenoid biosynthesis with the absence of Z-ISO in red algae [35]. This result may indicate other genes in red algae that can replace the gene for Z-ISO. However, there is a lack of evidence for this hypothesis.

### 3.3. Lycopene to α-Carotene and Derivatives

The cyclization of lycopene to carotenoid is catalyzed by lycopene cyclase, a group of enzymes, such as crtY, plant LCY; crtL, *cyanobacteria* LCY; crtYB, bifunctional fungal LCY [36]. Lycopene is converted to α-carotene or β-carotene by different enzymes, which is the most crucial process during the whole pathway as a branching point. The formation of α-carotene is catalyzed by enzymes LCYE, and LCYB catalyzes the formation of α-carotene. In this process, the end group on one side of lycopene starts to cyclize into δ-carotene. Then the cyclization of another end group is catalyzed by the enzyme LCYB, and the whole α-carotene is synthesized [37]. The LCY family exists among microalgae due to gene loss, gene duplication, or gene transfer [38]. 

Lutein is one of the derivatives of α-carotene. The formation involves two steps: α-carotene is converted to zeaxanthin or α-cryptoxanthin, and then zeaxanthin is converted to lutein, which are both catalyzed by P450-type enzymes ϵ-hydroxylase and β-hydroxylase [21]. Lutein was founded on the stigma of *E. gracilis* v. *fuscopunctata* [28]. Lutein is an essential pigment for the eyes, as a critical agent in the retina, even though it has a lower level in the lens [39].

### 3.4. Lycopene to β-Carotene and Derivatives

The biosynthesis of β-carotene involves two sequential steps catalyzed by the LCYB enzyme. Firstly, lycopene is converted to γ-carotene, andγ-carotene is converted to β-carotene [40]. Light-harvesting complex (LHC) has dual functions, collecting the light and transferring excitation energy into the reaction center [41]. LHC has four carotenoid binding sites, two lutein molecules, a neoxanthin molecule, and a carotenoid molecule. In LHC, carotenoids may play a part in light-harvesting and photoprotection [42]. β-carotene might have a protective and light-harvesting function in both RCs [4]. 

There are kinds of derivatives of β-carotene in algae and plants. This review clarifies some of those studied in detail in some microalgae. When β-carotene is synthesized, it is converted into zeaxanthin by the enzyme BCH in two steps of hydroxylase reactions, in the first step, β-carotene is converted into β-cryptoxanthin and then transformed into zeaxanthin [21]. Zeaxanthin could be found in the peripheral LHC of photosystem I in red algae, which could disperse the excess energy from excited chlorophylls [4]. The biosynthesis pathway of violaxanthin is known as xanthophyll cycle-I. During conditions of darkness or low light illumination, the synthetic process of violaxanthin is catalyzed by the enzyme ZEP, which needs two steps [43]. Zeaxanthin is converted to antheraxanthin, and in the next step, antheraxanthin is converted to violaxanthin. The enzyme ZEP catalyzes two steps with the epoxidation reaction of β-rings [44]. During high light illumination, the enzyme VDE catalyzed the violaxanthin transfer to zeaxanthin, which was activated by decreasing pH [45]. Violaxanthin has two different ways to form fucoxanthin. Firstly, violaxanthin is converted to neoxanthin in the next step, and neoxanthin is converted to fucoxanthin, in higher plants [21]. The enzyme NXS catalyzes the former reaction, and VDL has the same function [46]. However, there is no evidence of which enzyme could catalyze the neoxanthin to turn into fucoxanthin. In *Euglena*, we still have not found a trace of fucoxanthin. In another way, violaxanthin is converted to diadinoxanthin, then the diadinoxanthin is converted to fucoxanthin, in some microalgae [21]. The enzymes of this process are also unknown. In some algae, the diadinoxanthin could convert into diatoxanthin, catalyzed by the enzyme DDE. This pathway is called xanthophyll cycle-II [47]. Neoxanthin was found to help lessen the risk of lung cancer according to regulating the consumption of 9-(Z) neoxanthin [48]. 

Astaxanthin, a crucial and significant commercial product, has a large-scale application in different situations, such as in the food, cosmetics, and medical industries [49]. In some species of *Euglena*, we found the astaxanthin, such as *E. gracilis* [19], *E. sanguinea* [14], and *E. rubida* Mainx [15]. Scientists have proposed two different ways to synthesize this carotenoid. According to one way, β-carotene is used as a precursor for the biosynthesis pathway of canthaxanthin catalyzed by β-carotene ketolase (BKT). The canthaxanthin is converted to astaxanthin catalyzed by β-carotene hydroxylase (BCH). Similarly, zeaxanthin is converted to astaxanthin through adonixanthin by the enzyme BKT [21]. A small amount of β-carotene derivatives, such as loroxanthin, adonirubin, and adonoxanthin, have been found in *E**. sanguinea*. Another derivative-heteroxanthin was found in *E**. viridis* [50]. 

## 4. Genes of Carotenoids Biosynthesis

Without complete *Euglena* genome information, the full picture of carotenoid biosynthesis in this genus is still ambiguous. Minimal fragmented information was collected via single gene investigations. So far, no gene knock-off study has been performed using the CRISPR-Cas9 genome editing tool in *Euglena*, though this method was recently set up successfully [51].

CrtB (PSY) catalyzes two-molecules GGPP condense to a phytoene. *EgcrtB* could be induced by high-intensity light illumination(continuous illumination at 920 μmol m^−2^ s^−1^). RNA interference (RNAi) is used to silence *EgcrtB* expression in E. gracilis. After the treatment, the proliferation and chlorophyll and carotenoid content of *E. gracilis* were decreased apparently [29]. The paramylon granules were accumulated more in the cytoplasm of *EgcrtB*-suppressed cells compared with without *EgcrtB*-dsRNA cells [31]. 

Lycopene β-cyclase (LCY), such as LCYB, CrtL, CruA, CruP, and CrtY, are present in plants, eukaryotic algae, and bacteria. CruA and CruP were found in bacteria, and CrtL-type proteins are characterized in all plants [52]. EgLCY is encoded by an uncharacterized gene, which converts lycopene to β-carotene. After RNAi knock-down, the color, carotenoid content, density, and chlorophyll content of *E. gracilis* have changed. Mutants of *EgLCY* knock-down produced colorless cells with hypertrophic appearance and inhibited growth. They marked a decrease in carotenoid and chlorophyll content, suggesting that EgLCY is essential for synthesizing β-carotene and downstream carotenoids. Moreover, in *EgLCY* knock-down mutant cells, the APX activity was down-regulated, indicating a possible relationship between carotenoid biosynthesis and the ascorbate-glutathione cycle for ROS elimination [53]. 

The CYP97 family is divided into five subgroups, such as clans A, B, C, E, and F, from plants and eukaryotic algae. *EgCYP97H1*, a cytochrome P450(CYP)-type carotene hydroxylase gene, was found in *E. gracilis*, which converts β-carotene to β-cryptoxanthin as a β-carotene monohydroxylase. *EgCYP97H1* does not belong to the above subgroups but composes clan H independently. In *E. gracilis* mutant deficient in *EgCYP97H1*, total carotenoid content was decreased to 57% compared with control cells (84%) [54]. There is a strong correlation between the content of carotenoids and the transcriptional level of carotenogenesis genes in some microalgae, such as *H. pluvialis* [54], *Parachlorella kessleri* [55]. However, the transcriptional levels of the three genes, *CYP97H1*, *EgcrtB*, and *EgcrtE*, remain unchanged, indicating that carotenoid biosynthesis might be regulated at the post-transcriptional level in *E. gracilis*.

## 5. Cultivation Conditions and Carotenoids Accumulation

It is well known that environmental conditions induce and determine the composition and contents of carotenoids in microalgae and higher plants. Microalgae can overproduce lipids or carotenoids under stress conditions such as high salt, high light, or nutrient limitation [56,57]. For instance, when salt concentration was increased from 4 to 9%, the β-carotene yield of *Dunaliella salina* was increased by 30-fold [58]. Cells of the green microalga *H. pluvialis* were induced to accumulate the ketocarotenoid pigment, astaxanthin, under high light (170 μmol m^−2^ s^−1^), phosphate starvation, and salt stress (NaCl 0.8%) [59]. In *Euglena*, some investigations about environmental stresses such as oxidative stress, heavy metals, nutrient starvation, and different cultivation conditions were conducted to reveal the carotenoid accumulation in *Euglena*. 

### 5.1. Environmental Stresses

Anthracene is a three-ring, low molecular weight PAH, considered a harmful contaminant to the environment, especially the water environment [60]. Anthracene was produced by volcanic eruptions, crude oil spills, automobile exhausts, and so on [61]. Anthracene substantially affects microorganisms due to ROS formation [62]. When *E**. agilis* was exposed to anthracene for 96 h, the cell density was reduced due to the dissipation of the electrical potential and pH gradient [63]. What is more, the content of carotenoids was significantly affected by the anthracene (>0.625 mg/L), which was reduced to 17% (Table 2). At the highest concentration of anthracene (15 mg/L), the content of carotenoids was reduced to 49% [64]. Carotenoids as antioxidants act to reduce the ROS reaction, which might be why it would be reduced.

Arsenic is one of the most poisonous pollutants produced by fossil fuel, glass production, and agricultural applications [70]. Its toxicity is caused by binding to cellular membranes and arsenic substitutes for phosphate or choline head of phospholipid [71]. At the highest dose of As-III, the total carotenoids ranged from 1.8 μg/mL to 0.4 μg/mL. In addition, chlorophylls were more sensitive than carotenoids; therefore, carotenoids could protect the cells from oxidative stress induced by arsenic [67].

UV-B radiation could affect the motility and orientation of *E. gracilis*. Even though short-term exposure to UV-B radiation would decrease the percentage of motile cells and reduce the swimming velocity of the microorganisms [72]. After 16 h of continuous UV-B irradiation, carotenoids were increased; this phenomenon might relate to the function of photoprotective of astaxanthin [5]. 

Nitrogen (N) availability could regulate the partition of carbon (C) into starch or lipids [73]. In N-limited growth, C turns into non-N-containing compounds, especially neutral lipids, carbohydrates, and carotenoids. However, carotenoid production decreased during short-term exposure to N deprivation in *E. gracilis* [74]. The concentrations of carotenoids in low N treatment were lower than in high N treatment [75]. As we all know, carotenoids are N-free compounds. These results may be caused bythe synthesis of pigment-protein complexes in the thylakoid membrane; as N. Boussiba suggested the response to N and P limitations differs depending on the kind of carotenoid. For instance, under N and P limitations, astaxanthin was increased in *H**. pluvialis*, but lutein and β-carotene were decreased [76].

### 5.2. Cultivation Conditions

*Euglena* can grow under autotrophic, heterotrophic, or mixotrophic conditions. Under the mixotrophic medium with glucose as C source and (NH_4_)_2_SO_4_ as N source, with pH from 3.3 to 1.8, the density and growth rate of cells is higher than that in autotrophic or heterotrophic with the same C and N sources [66]. This result reveals that the mixotrophic condition was the best way for the growth of *Euglena*, followed by the heterotrophic. The yield of mixotrophic is higher than that in heterotrophic because, firstly, the formation of ATP through photochemical reactions; secondly, in autotrophic culture, ATP cannot be used for the cell metabolism due to the absence of a CO_2_-concentrating mechanism [77]. In the mixotrophic condition, the contents of chlorophylls and carotenoids were gradually increased with a culture proceeding. 

Light is an essential parameter for a photosynthetic organism. The dark growth cells exposed to light would increase the content of carotenoids. After cultivation in the dark for ten days, cells were exposed to light, and the total content of carotenoids was from 0.25 pg/cell to 0.80 pg/cell. With the light density ranging from 300 lux to 4000 lux, the content of carotenoids increased gradually [16]. Pre-radiation with low-density red or blue light in *E. gracilis* cells could increase the accumulation of β-carotene, which plays a role in alleviating light-induced stress in the early stage. In order to increase the accumulation of kinds of carotenoids in *E. gracilis*, different intensities of red or blue light are required [69].

Temperature is one of the most striking external factors for the production of carotenoids. Several studies have proved the optimum temperature range for *E. gracilis* was 25 °C to 33 °C, similar to green algae [78]. A temperature of around 22 °C is considered a “cold stress” threshold for *E. gracilis*. The phytoene desaturase genes (*crtP1* and *crtP2*) and phytoene synthase gene(*crtB*) respond to cold–intense light treatment. In addition, cold stress can enhance microalgal susceptibility to light-induced stress [33]. 

### 5.3. Extraction and Analysis

There are several kinds of organic solvents for carotenoid extraction of microalgae, including ethanol, acetone, a mixture of ethanol and acetone, hexane, and so on [54,79,80,81]. According to the chemical stability, solubility, and polarity of compounds, the selection of solvent and pressure of extraction is specific [82]. Grounding with glass beads and ultrasound-assisted solvent extraction play a role in carotenoid extraction [83]. Moreover, treatment with mild acids could increase the extraction yield due to their capacity to disrupt the cells. The use of strong acids has an opposite effect [84]. The disadvantages of traditional methods are that they are slow to process and produce toxic organic solvents. The novel extraction methods are environmentally friendly, efficient, and sustainable, and could extract the pigments without damaging their activity; methods such as pressurized-liquid extraction (PLE), microwave-assisted extraction (MA), ultrasound-assisted extraction (UAE), enzyme-assisted extraction (CAE), subcritical water extraction (SWE), and supercritical fluid extraction (SCFE) [81]. Using CO_2_ as a solvent in supercritical fluid extraction makes it one of the most promising methods in industrial application. Supercritical fluid extraction could prevent solvent traces in the final products [82]. In addition, CO_2_ could be recovered safely and economically. 

The analysis methods contain HPLC, Ultra Performance Liquid Chromatography (UPLC) [68], Gas Chromatographic Mass Spectrometer (GC/MS), Ultraviolet-visible spectroscopy (UV/Vis) quantitative and qualitative analysis [85]. HPLC and UV/Vis methods have been widely used in recent years. The Lomakin–Schiebe formula is the basis of UV/Vis, which has been widely used. The absorption peak of carotenoids is 470nm [86]. The esterification of carotenoids cannot be identified by those traditional tools due to the small spectral shifts. The method of HPLC/APCI-MS could display the composition of the acyl chains of monoesters and diesters of carotenoids [81]. Using a syringe filter to filter the extracts is also beneficial for analysis.

## 6. Applications

Carotenoids, as natural pigments, have essential functions in life, for both plants and animals, especially humans. According to light-harvesting, excess light dissipation, photosynthesis protection, and so on, carotenoids have a wide range of applications in the world market. Antioxidant functions of carotenoids are based on mechanisms such as quenching free radicals, mitigating damage from reactive oxidant species, and hindering lipid peroxidation. Many studies have suggested that carotenoids protect humans from the risk of cancers, which gives them vast potential. Their dietary functions decrease the risk of breast, cervical, vaginal, and colorectal cancers as well as cardiovascular and eye diseases. In addition, carotenoids could also be used in cosmetic industries, food industries, and animal feed, as shown in Table 3.

Due to the possibility of reducing the adverse effect of ROS, many studies have claimed that carotenoids have a protective effect on the development of chronic and degenerative diseases [9]. The microalgae biomass could significantly increase the antioxidant enzymes in the plasma and liver of mice, which means carotenoids could scavenge free radicals to protect cells [97]. Patients with HIV have a meager lymphocyte ratio of CD4 to CD8; a study showed that high doses of β-carotene could increase this ratio. Moreover, lycopene probably has the greatest anticancer capacity compared to other carotenoids [98]. The 60% to 80% of total carotenoids in testes and adrenal glands are take-ups placed by lycopene [99]. Lycopene has been proven to inhibit tumor cells in vivo and in vitro. Regular consumption of 9-(Z) neoxanthin might also reduce the risk of lung cancer [48].

Carotenoids have another function of protecting the low-density lipoprotein (LDL)from oxidation [100], which confers the carotenoid’s antiatherogenic properties [90]. Several studies suggested that with β-carotene supplementation in diet, total lipids, cholesterol, and triglycerides would be decreased in mammals [101]. A-carotene was proposed as a potential marker for human atherosclerosis because low levels have an inversely correlated prevalence of coronary artery disease and arterial plaque formation. Additionally, high lycopene levels have been inversely linked to the risk of suffering from myocardial infarction. The tendency to suffer from myocardial is related to the low intake of lutein [9]. 

Lutein and zeaxanthin maintain the normal visual function of the human eye and the yellowing [102]. They could absorb excess blue light to protect the eye macula from being damaged by blue light due to their absorption band with a peak of 450 nm [103]. According to their antioxidant properties, carotenoids could protect the macula from photochemical reactions [104]. 

Lycopene could protect DNA from oxidative damage in vitro. With lycopene (5 μmol/L) preincubated with human spermatozoa, sperm motility and DNA integrity have been shown to be well protected. After taking astaxanthin capsules, skin wrinkles, elasticity, age spots, and corneocytes significantly improve after eight weeks [105]. This result may reveal the potential of astaxanthin in the cosmetic industry. 

## 7. Conclusions

Carotenoids are widely used in different situations, such as in food, animal feed, cosmetics, and especially as an anticancer treatment in medical industries. We discuss the distribution and structure of the main carotenoids in species and strains of *Euglena*. The biosynthesis of carotenoids in *Euglena* still has no clear evidence. Thus, the carotenoid composition inferred the entire pathway according to the pathways studied in other microalgae. With an incomplete genome sequence and immature genome editing tools, gene mining for carotenoid biosynthesis is still focused on a couple of genes using RNAi techniques. In this review, *EgCYP97H1* was found only in *E. gracilis*, which means it could be a unique biosynthesis pathway that existed in *E. gracilis*. Not only could environmental stresses affect the growth and carotenoid accumulation of *Euglena*, but so could cultivation parameters such as light, temperature, and carbon/nitrogen sources. Increasing the knowledge of carotenoids in *Euglena* has vast potential for commercial and social value. Further complements of genome assembly, systems biology study to discover target genes, mature genetic engineering, and genome editing tools to explore the gene functions and employ synthetic biology will help advance *E.gracilis* as a valuable model microorganism for basic research and industrial product and development. 

## Figures and Tables

**Figure 1 marinedrugs-20-00496-f001:**
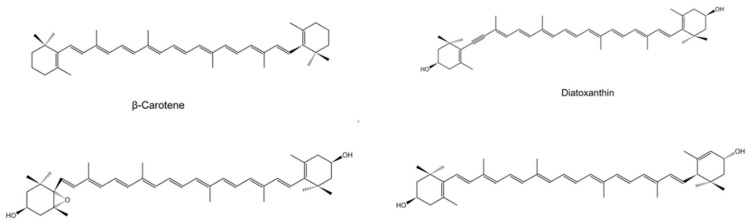
Structures of major carotenoids by *Euglena*.

**Figure 2 marinedrugs-20-00496-f002:**
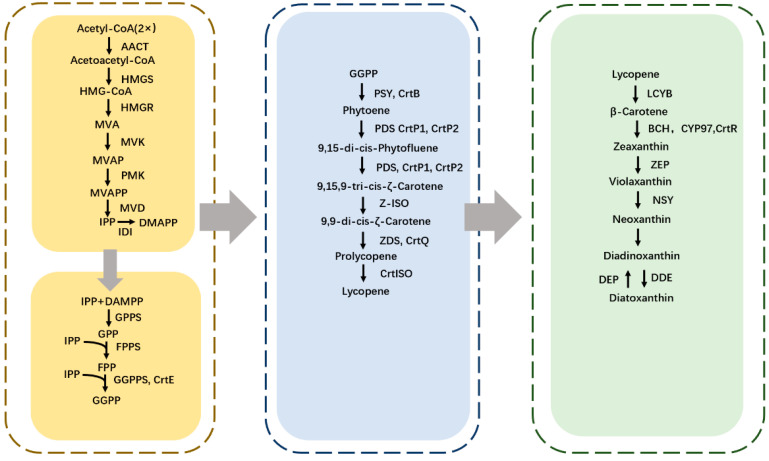
Biosynthesis pathway of carotenoids in *Euglena*.

**Table 1 marinedrugs-20-00496-t001:** Comparison of composition and contents of carotenoids among strains of *Euglena*.

Strains	Pigments	Common Pigment	Analysis Method	References
*E. sanguine*	Diatoxanthin; Lutein; β-carotene	β-carotene	HPLC	[14]
*E. rubida*	Astaxanthin; Mutatoxanthin; β-carotene	β-carotene	HPLC	[15]
*E. gracilis*	Diadinoxanthin; β-carotene; Diatoxanthin; Neoxanthin	β-carotene	HPLC	[12]
SM-bleached of *E. gracilis*	Euglenanon; Antheraxanthin; β-carotene	β-carotene	HPLC	[16]
Pressure-bleached of *E. gracilis*	Phytofluene; ξ-carotene; β-carotene, β-zeacarotene	β-carotene	TLC	[17]
*EgcrtB*-suppressed *E. gracilis*	β-carotene; Astaxanthin; Zeaxanthin; Canthaxan-thin	β-carotene	HPLC	[12]
SM-bleached of *E. gracilis* Z	β-carotene; Astaxanthin; Zeaxanthin; Canthaxan-thin	β-carotene	HPLC	[12]

**Table 2 marinedrugs-20-00496-t002:** Effects of environmental stresses, cultivation methods on the production of carotenoids by *Euglena*.

Strains	Parameters	Treatment/Time	Result	References
*E. sanguinea*	Field	UV-B; 16 h	Increased	[5]
*E. gracilis* (NIES-48)	L:D 12:12h;T:25 °C; M: CM	HAM; 25 days	Increased	[65]
*E. gracilis* (CCAP 1224/5Z)	L:D 16:8T:25 °C; M: Hutner	(NH_4_)_2_SO_4_; 25 days	Increased	[66]
*E. gracilis* Z	L:D 24:0;T:24 °C; M: Checcucciet	As_2_O_3_; 7 days	0.4~1.8 μg/mL	[67]
*E. gracilis* Z	L:D 24:0; T:25 °C; M: CM	20 °C; 7 days	Reduced	[68]
*E. gracilis*	L:D 16:8; T:25 °C; M: mineral medium;	Anthracene; 96 h	Reduced	[64]
*E. gracilis* Z	L:D NS; T:28 °C; M: Oda	Autotrophic; mixotrophic &heterotrophic cultures; 125 h	mixotrophic > autotrophic > heterotrophic	[16]
*E.gracilis* Z	L:D 12:12; T:25 °C; M: CM	Low identity blue and red light; 24 h	Increased	[69]
*E. gracilis* Z	L:D NS;T: 27 °C; M: Hunter	Light; 10 days	Darkness < light	[15]

L: light; D: dark; T: temperature; M: medium; NS: not specified.

**Table 3 marinedrugs-20-00496-t003:** Summary studies of biotechnological applications and results related to the medicine of carotenoids from *Euglena*.

Strains	Carotenoids	Applications	References
*Euglena*	β-carotene	Natural colorant and additive in the cosmetic and food industries; Anticancer;	[87,88]
*E. sanguine,* *E. rubida*	Astaxanthin	Dietary supplement; Protecting neurons; Antiobese; Softening; Diabetes effects; Hepatoprotector; Anticancer	[89,90,91,92]
*E. gracilis* Z	Echinenone	Edible orange pigments; Antioxidants; Cosmetic industries	[93]
*E. sanguine*	Adonixanthin	Anti-inflammatory and antioxidant	[94]
*E. sanguine,* *E. rubida,* *E. gracilis*	Neoxanthin	Antioxidant	[95]
*E. sanguine*	Diatoxanthin	Anti-inflammatory	[96]

## Data Availability

No new data were created or analyzed in this study. Data sharing is not applicable to this article.

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
