# Peer review of "Carotenoids Biosynthesis, Accumulation, and Applications of a Model Microalga Euglenagracilis"

_marinedrugs, 2022, doi:10.3390/md20080496_

Round 1

Reviewer 1 Report

The present study is about the biosynthesis, accumulation, and functions of carotenoids from a model microalga Euglena gracilis and its potential applications in the field of therapy preventing carcinogenesis, cosmetic industries, food industries, and animal feed. There are few studies on the carotenoids of the flagellated alga, Euglena gracilis. For this reason, this article is interesting and novel as a well-designed review.

 Line 9: This sentence should be revised. The carotenoids cannot be classified as only isoprenoids. The detail should be given to make it accurate. As a suggestion:

“The carotenoids belong to the isoprenoids and their basic structure is made up of eight isoprene units, resulting in a C40 backbone”

 Line 302: 5.2. Cultivation conditions

I strongly recommend the authors add another subsection (5.3. Extraction and analysis of carotenoids from Euglena) and discuss the extraction and analysis of carotenoids from Euglena strains. According to me, the only missing part in this MS is this part.

Author Response

Reviewer 1 | 08 Jul 2022 | 10:17

#1

The present study is about the biosynthesis, accumulation, and functions of carotenoids from a model microalga Euglena gracilis and its potential applications in the field of therapy preventing carcinogenesis, cosmetic industries, food industries, and animal feed. There are few studies on the carotenoids of the flagellated alga, Euglena gracilis. For this reason, this article is interesting and novel as a well-designed review.
1. Line 9: This sentence should be revised. The carotenoids cannot be classified as only isoprenoids. The detail should be given to make it accurate. As a suggestion:

The carotenoids belong to the isoprenoids and their basic structure is made up of eight isoprene units, resulting in a C40 backbone”

Response to Reviewer: The sentence is revised as “The carotenoids, including lycopene, lutein, astaxanthin, and zeaxanthin belong to the isoprenoids, whose basic structure is made up of eight isoprene units, resulting in a C40 backbone, though some of them only trace components in Euglena.”

  1. Line 302: 5.2. Cultivation conditions.

I strongly recommend the authors add another subsection (5.3. Extraction and analysis of carotenoids from Euglena) and discuss the extraction and analysis of carotenoids from Euglena strains. According to me, the only missing part in this MS is this part.

Response to Reviewer: Thanks for your suggestions; we added a new chapter-5.3. extraction and analysis.

“There are kinds of organic solvents for carotenoids extraction of microalgae, including ethanol, acetone, a mixture of ethanol and acetone, hexane and so on [80-83]. According to the chemical stability, solubility, and polarity of compounds, the selection of solvent and pressure of extraction is specific [84]. Grounding with glass beads and ultrasound-assisted solvent extraction play a role in carotenoid extraction [85]. Moreover, the treatment of mild acids could increase the extraction yield due to the capacity to disrupt the cells. The use of strong acids has an opposite affecting [86]. The disadvantages of traditional methods are slow process and toxic organic solvent. The novel extraction methods are environmentally friendly, efficient and sustainable, could extract the pigments without damaging their activity. Such as, pressurized-liquid extraction (PLE), microwave-assisted extraction (MA), ultrasound-assisted extraction (UAE), enzyme-assisted extraction (CAE), subcritical water extraction (SWE), and supercritical fluid extraction (SCFE) [83]. Using CO2 as a solvent in supercritical fluid extraction makes it one of the most promising methods in industrial application. Supercritical fluid extraction could prevent solvent traces in the final products [84]. In addition, CO2 could be recovered safely and economically.

The analysis methods contain High-Performance Liquid Chromatography (HPLC), Ultra Performance Liquid Chromatography (UPLC) [69], Gas Chromatographic Mass Spectrometer (GC/MS), Ultraviolet-visible spectroscopy (UV/Vis) quantitative and qualitative analysis [87]. HPLC and UV/Vis methods have been widely used in recent years. The Lomakin-Schiebe formula is the basis of UV/Vis, which has been widely used. The absorption peak of carotenoids is 470nm [88]. The esterification of carotenoids cannot be identified by those traditional tools due to the small spectral shifts. The method of HPLC/APCI-MS could display the composition of the acyl chains of monoesters and diesters of carotenoids [83]. Using of syringe filter to filter the extracts is also beneficial for analysis.”

Please check the revised ms for more details.

Reviewer 2 Report

The submitted manuscript reviewed recent progress in Euglena’s carotenoid biosynthetic pathway, functions, and their applications. It is timely, most of the recent papers were well cited, and the structure of the review is excellent.  Some important and essential original papers and reviews should be cited and revied the manuscript to deepen the understanding of many readers.

(1)  The following review papers described by two research groups will help the authors make the submitted manuscript more effective.

Tamaki et al., 2020, Diverse Biosynthetic Pathways and Protective Functions against Environmental Stress of Antioxidants in Microalgae, https://doi.org/10.3390/plants10061250

Kato and Shinomura, 2020, Light dependent accumulation of β-carotene enhances photo-acclimation of Euglena gracilis, https://doi.org/10.1007/978-3-030-50971-2_4

(2) In Abstract and Introduction, some statements should be rewritten appropriately.

- Line 9-10: some carotenoids described are not major component in Euglena, so I am afraid that they are misleading to the reader.

- Line 39-40: “There is still no clear evidence” is an overstatement.

- Line 45-47: Please read and cite the following paper and rewrite appropriately.

Kato et al., 2020, Carotenoids in the eyespot apparatus are required for triggering phototaxis in Euglena gracilis, https://doi.org/10.1111/tpj.14576

(3) “2. Composition and structure …”and Table 1 is required some modification by referring the following papers. Kato et al, 2016 showed major carotenoid components different from the reference [11]. In addition, Tamaki et al., 2020 showed the major carotenoid components of SM-bleached of E. gracilis different from the reference [15].

Kato et al., 2016, Identification and functional analysis of the geranylgeranyl pyrophosphate synthase gene (crtE) and phytoene synthase gene (crtB) for carotenoid biosynthesis in Euglena gracilis, DOI 10.1186/s12870-015-0698-8

Tamaki et al., 2020, Carotenoid accumulation in the eyespot apparatus required for phototaxis is independent of chloroplast development in Euglena gracilis, https://doi.org/10.1016/j.plantsci.2020.110564

(4) “3. 1. IDI to GGPP” needs to cite Kato et al., 2016 that is the exact paper related to the sub-title.

(5) “3.2. GGPP to Lycopene” should cite reference [45] and the following paper to the appropriate places.

Kato et al., 2019, Low temperature stress alters the expression of phytoene desaturase genes (crtP1 and crtP2) and ζ-carotene desaturase gene (crtQ) of Euglena gracilis and the cellular carotenoid content, DOI 10.1093/pcp/pcy208

Sugiyama et al., 2019, Oxygenic Phototrophs Need ζ-Carotene Isomerase (Z-ISO) for Carotene Synthesis: Functional Analysis in Arthrospira and Euglena, DOI 10.1093/pcp/pcz192

(6) The papers shown should be appropriately cited to “4. Genes of carotenoid …”.

(7) “5.1. Environmental stress” , Table 2 and “5.2. Cultivation condition“ lacked the following important paper.

Tanno et al., 2020, Light dependent accumulation of β-carotene enhances photo-acclimation of Euglena gracilis, https://doi.org/10.1016/j.jphotobiol.2020.111950

(8) The title of the submitted paper included “function”, but corresponding section was not shown. In “6. Application”, some function was described, but is not enough to review. Especially, one of the important functions of carotenoids for phototaxis was just slightly described in Introduction.

Author Response

Reviewer 2 | 14 Jul 2022 | 16:19

#1

The submitted manuscript reviewed recent progress in Euglena’s carotenoid biosynthetic pathway, functions, and their applications. It is timely, most of the recent papers were well cited, and the structure of the review is excellent.  Some important and essential original papers and reviews should be cited and revied the manuscript to deepen the understanding of many readers.

1.The following review papers described by two research groups will help the authors make the submitted manuscript more effective.

Tamaki et al., 2020, Diverse Biosynthetic Pathways and Protective Functions against Environmental Stress of Antioxidants in Microalgae, https://doi.org/10.3390/plants10061250

Kato and Shinomura, 2020, Light dependent accumulation of β-carotene enhances photo-acclimation of Euglena gracilis, https://doi.org/10.1007/978-3-030-50971-2_4

Response to Reviewer: According to papers you suggested, we made some revisions. Firstly, we modified the figure 2-Biosynthesis pathway of carotenoids in Euglena. In addition, some statements also were modified. For example, line 92 in original “However, there is no evidence to show astaxanthin in E. gracilis”, which was revised as “Astaxanthin are also found in non-photosynthetic mutants of E. gracilis, such as suggesting the synthesis pathway of astaxanthin might present in E. gracilis.”

  1. In Abstract and Introduction, some statements should be rewritten appropriately.

- Line 9-10: some carotenoids described are not major component in Euglena, so I am afraid that they are misleading to the reader.

Response to Reviewer: Thanks for your suggestions. We modified this sentence as “The carotenoids, including lycopene, lutein, astaxanthin, and zeaxanthin belong to the isoprenoids, whose basic structure is made up of eight isoprene units, resulting in a C40 backbone, though some of them only trace components in Euglena.”

- Line 39-40: “There is still no clear evidence” is an overstatement.

Response to Reviewer: The sentence was revised as “According to recent studies, the synthesis pathway of carotenoids has not been fully identified in Euglena.”

- Line 45-47: Please read and cite the following paper and rewrite appropriately.

Kato et al., 2020, Carotenoids in the eyespot apparatus are required for triggering phototaxis in Euglena gracilis, https://doi.org/10.1111/tpj.14576

Response to Reviewer: The sentence is revised in lines 47-51 as “There are two major hypothesis about the function of eyespot in Euglena including, i)as a device of shading, which could help cell to change its orientation when the direction of light changes; ii) as a device of primary photoreceptor. According to the study of Kato in 2020 indicates that eyespot is essential for the phototaxis of Euglenophyta and the carotenoids in eyespot apparatus performs an important role in photoreception.”

  1. “2. Composition and structure …”and Table 1 is required some modification by referring the following papers. Kato et al, 2016 showed major carotenoid components different from the reference [11]. In addition, Tamaki et al., 2020 showed the major carotenoid components of SM-bleached of E. gracilis different from the reference [15].”

Kato et al., 2016, Identification and functional analysis of the geranylgeranyl pyrophosphate synthase gene (crtE) and phytoene synthase gene (crtB) for carotenoid biosynthesis in Euglena gracilis, DOI 10.1186/s12870-015-0698-8

Tamaki et al., 2020, Carotenoid accumulation in the eyespot apparatus required for phototaxis is independent of chloroplast development in Euglena gracilis, https://doi.org/10.1016/j.plantsci.2020.110564

Response to Reviewer: Lines 71-73 in original “Carotenoids were found in E. gracilis, antheraxanthin, β-Carotene, neoxanthin, γ-carotene, hydroxyechinenone, euglenanone, and echinenone, and the last three ca-rotenoids may be the pigment of the stigma due to their orange color”

This sentence is revised as in lines 75-78: “The major carotenoids were found in E. gracilis, including diadinoxanthin, diatoxanthin, β-carotene, and neoxanthin, shown in table 1. The carotenoids in the stigma of Euglena were diadinoxanthin, diatoxanthin and β-carotene, which are related to phototaxis [12].”

In addition, the composition of carotenoids in E. gracilis in the table has also been modified.

  1. “3. 1. IDI to GGPP” needs to cite Kato et al., 2016 that is the exact paper related to the sub-title.”

Response to Reviewer: In order to cite the paper written by Kato in 2016, adding a sentence in Line 139 “EgcrtE is a functional gene that encodes GGPP synthase in E. gracilis [29].”

  1. “3.2. GGPP to Lycopene” should cite reference [45] and the following paper to the appropriate places.”

Kato et al., 2019, Low temperature stress alters the expression of phytoene desaturase genes (crtP1 and crtP2) and ζ-carotene desaturase gene (crtQ) of Euglena gracilis and the cellular carotenoid content, DOI 10.1093/pcp/pcy208

Sugiyama et al., 2019, Oxygenic Phototrophs Need ζ-Carotene Isomerase (Z-ISO) for Carotene Synthesis: Functional Analysis in Arthrospira and Euglena, DOI 10.1093/pcp/pcz192

Response to Reviewer: Lines 147-149 “PSY has a significant effect on the carotenoids biosynthesis pathway due to its rate-limiting and total flux controlling, which decides the content of carotenoids [21]” two sentences were added here “EgcrtB was confirmed to encode phytoene synthase in E. gracilis. The content of carotenoids has a significant decrease in EgcrtB-suppressed cells [31].

Lines 154-155 “In the first two-step, phytoene is converted to 9,15,9-tri-cis-ξ-carotene all by the PDS enzyme”, a sentence was added here “The phytoene desaturase genes (EgcrtP1 and EgcrtP2) were identified by kato in 2018, and both could encode the phytoene desaturase [33].”

Lines 157-159 “The 9,15,9-tri-cis-ξ-carotene then is converted to 9,9-dis-cis-ξ-carotene and then to prolycopene, and final to lycopene are catalyzed by Z-ISO, ZDS, and CrtlSO, respectively [19]” three sentences were added here “ζ-carotene isomerase gene (Z-ISO) in E. gracilis was related to Z-ISO of chlorophyta. What is more, Z-ISO is essential for oxygenic phototrophs, but not be found in anoxy-genic phototrophs [33]. ζ-carotene desaturase gene (EgcrtQ) in E. gracilis has been proven to be a clade of ZDS [34].”

  1. The papers shown should be appropriately cited to “4. Genes of carotenoid …”.

Response to Reviewer: Thanks for your suggestion, and we have modified some statements in lines 232-237.

  1. 5.1. Environmental stress”, Table 2 and “5.2. Cultivation condition “lacked the following important paper.”

Response to Reviewer: Line 321-324 add sentences “Pre-radiation with low-density red or bule light in E. gracilis cells could increase the accumulation of β-carotene, which plays a role in alleviating light-induced stress in the early stage. In order to increase the accumulation of kinds of carotenoids in E. gracilis, different intensities of red or blue light are required [70].”

Strains

Parameters

Treatment/Time

Result

Reference

Euglena gracilis Klebs (strain Z)

L: D12:12

T:25℃

M:CM

Low identity blue and red light; 24h

Increased

[70]

  1. The title of the submitted paper included “function”, but corresponding section was not shown. In “6. Application”, some function was described, but is not enough to review. Especially, one of the important functions of carotenoids for phototaxis was just slightly described in Introduction.

Response to Reviewer: In this paper, we pay more attention in the application of carotenoids, rather than functions of them. Should we change original title of this manuscript, which means delete the word “function”. And the chapter six-Application aims to make a simple statement about application of carotenoids to arouse interesting of people, who wants to use carotenoid to make kinds of products in different field.